# Neurofascin and Kv7.3 are delivered to somatic and axon terminal surface membranes en route to the axon initial segment

Aniket Ghosh, Elise LV Malavasi, Diane L Sherman, Peter J Brophy*

Centre for Discovery Brain Sciences, University of Edinburgh, Edinburgh, United Kingdom

**Abstract** Ion channel complexes promote action potential initiation at the mammalian axon initial segment (AIS), and modulation of AIS size by recruitment or loss of proteins can influence neuron excitability. Although endocytosis contributes to AIS turnover, how membrane proteins traffic to this proximal axonal domain is incompletely understood. Neurofascin186 (Nfasc186) has an essential role in stabilising the AIS complex to the proximal axon, and the AIS channel protein Kv7.3 regulates neuron excitability. Therefore, we have studied how these proteins reach the AIS. Vesicles transport Nfasc186 to the soma and axon terminal where they fuse with the neuronal plasma membrane. Nfasc186 is highly mobile after insertion in the axonal membrane and diffuses bidirectionally until immobilised at the AIS through its interaction with AnkyrinG. Kv7.3 is similarly recruited to the AIS. This study reveals how key proteins are delivered to the AIS and thereby how they may contribute to its functional plasticity.

*For correspondence:
peter.brophy@ed.ac.uk

**Competing interests:** The authors declare that no competing interests exist.

## Introduction

Neurons are highly polarised cells with functionally distinct membrane domains. The axon initial segment (AIS) is located at the proximal part of the axon where the high density of voltage-gated sodium channels (Nav) promotes the initiation and propagation of nerve impulses (*Leterrier, 2018*; *Palay et al., 1968*). After AIS assembly during development, this domain can retain a degree of plasticity such that changes in its size and length can influence neuronal excitability in the mature nervous system (*Grubb et al., 2011*; *Petersen et al., 2017*; *Kuba, 2012*). This morphological plasticity reflects the ability of the AIS to change the amount of its constituent proteins rather than their density (*Evans et al., 2015*). However, whether membrane proteins are exclusively inserted directly into the AIS as AnkG/membrane protein complexes (*Leterrier et al., 2017*), are concentrated at the AIS by selective endocytosis, or primarily arrive by lateral diffusion in the membrane from other insertion sites, or indeed whether all three mechanisms apply remains uncertain (*Akin et al., 2015*; *Barry et al., 2014*; *Boiko et al., 2007*; *Brachet et al., 2010*; *Fréal et al., 2019*; *Hamdan et al., 2020*; *Leterrier et al., 2017*; *Nakada et al., 2003*; *Torii et al., 2020*; *Winckler et al., 1999*; *Yap et al., 2012*; *Zonta et al., 2011*). Hence, determining the pathways by which membrane proteins are delivered to the AIS is not only important for understanding nervous system development, but may also shed light on how excitability is modulated in the mature neuron.

Neurofascin186 (Nfasc186) is a transmembrane protein with an essential role in maintaining the intactness of the AIS complex and in restricting AIS proteins to this specialized domain (*Alpizar et al., 2019*; *Boiko et al., 2007*; *Jenkins and Bennett, 2001*; *Zonta et al., 2011*). Deletion of Nfasc186 in culture and in vivo causes the disintegration of the AIS with the loss of Nav, AnkG,

βIV-Spectrin and Nr-CAM; the consequent disordered electrophysiology impairs motor learning (*Alpizar et al., 2019*; *Zonta et al., 2011*).

In this study, we show that vesicles transport Nfasc186 to two spatially distinct locations in cortical neurons, the cell soma and the axon terminus, where they fuse with the neuronal membrane. Analysis by fluorescence recovery after photobleaching (FRAP) combined with fluorescence loss in photobleaching (FLIP) shows that Nfasc186 is highly mobile in the neuronal membrane and that lateral diffusion in the axon, both proximally and distally to the AIS from the soma and axon terminal respectively, is primarily responsible for Nfasc186 delivery to the AIS. Unlike Nav1.6, direct fusion of transport vesicles at the proximal axon does not contribute to the accumulation of Nfasc186 at the AIS (*Akin et al., 2015*). Interaction with AnkG immobilises Nfasc186 at the AIS but is unnecessary for the incorporation of the protein into the axonal membrane. Kv7.3 also interacts with AnkG (*Pan et al., 2006*; *Zhang et al., 1998*) and follows a similar route to the AIS (*Devaux et al., 2004*; *Pan et al., 2006*; *Rasmussen et al., 2007*; *Shah et al., 2008*).

## Results and discussion

### Nfasc186 is inserted into the neuronal membrane at the soma and axon terminus

Nfasc186 is transported in vesicles generated in the secretory pathway by microtubule-based fast axonal transport, which is probably Kinesin 1-dependent (*Barry et al., 2014*; *Bekku and Salzer, 2020*; *Fréal et al., 2019* ; *Ichinose et al., 2019*; *Thetiot et al., 2020*). In order to image the pathway by which these vesicles reach the neuronal plasma membrane, we expressed super-ecliptic pHluorin (SEP) fused to the extracellular domain of full-length Nfasc186 in cortical neurons. SEP is a pH-sensitive GFP-derivative that allows selective imaging of Nfasc186 expressed at the cell surface (*Ashby et al., 2004*; *Ashby et al., 2006*; *Hildick et al., 2012*; *Makino and Malinow, 2009*; *Martin et al., 2008*; *Wilkinson et al., 2014*).

First, we asked if SEP-Nfasc186 is accumulated at the AIS like endogenous Neurofascin. Enrichment of the fusion protein at the AIS relative to the soma or distal axon was not significantly different from that of endogenous neuronal Nfasc186, either on a wild type (WT) or a Neurofascin-null background (*Figure 1A and B*). We then wished to identify the earliest stages of its journey to the AIS. Hence, neurons were transfected at DIV 2, and imaged the next day prior to AIS formation. SEP-Nfasc186 was strongly expressed at the cell surface of the soma and axon terminal (*Figure 1C*). Coexpression of KHC560-halo confirmed the axon terminal as a primary location of SEP-Nfasc186 accumulation at the cell surface (*Figure 1C*; *Twelvetrees et al., 2016*). A line scan of SEP-Nfasc186 signal intensity at the cell soma, axon and axon terminal of the neuron in the upper panel of *Figure 1C* showed that fluorescence was readily detectable in the axonal membrane relative to background (*Figure 1D*). Neither the absence of the over-expressed Kinesin nor reduced Nfasc186 expression on a Neurofascin-null background influenced the localisation of SEP-Nfasc186 (*Figure 1—figure supplement 1A and B*), and immunostaining using an antibody against an extracellular domain of the endogenous protein revealed Nfasc186 at the membrane surface of the soma and the axon terminus as found for SEP-Nfasc186 (*Figure 1—figure supplement 1C*). We concluded that SEP-Nfasc186 is a suitable proxy for assessing the localisation of endogenous neuronal Neurofascin.

Live imaging revealed transient increases in the fluorescent intensity of SEP-Nfasc186 at the surface of the axon terminal suggesting possible exocytotic fusion events. (*Figure 2A*). Total internal reflection fluorescence (TIRF) microscopy can reveal the sites of exocytotic fusion of vesicles that transport SEP-fusion proteins in neurons (*Li et al., 2012*). TIRF analysis showed that surface delivery of SEP-Nfasc186 is particularly active at the cell body and axon terminus, but not at the axon itself (*Figure 2B and C*; *Video 1*). The periodic actin/spectrin axonal cytoskeleton may play a role in limiting exocytotic events in the axon (*Leterrier, 2018*).

### Lateral diffusion of Nfasc186 in the axonal membrane

In order to ask if Nfasc186 can move retrogradely from the axon terminal in the axon membrane or if it remains at the axon terminal and is simply retrogradely transported back to the soma by vesicular transport, we adopted two approaches. First, neurons were transfected at DIV 3–4 and the lateral mobility of SEP-Nfasc186 in the axonal membrane was analysed ~16 hr later after subjecting a region

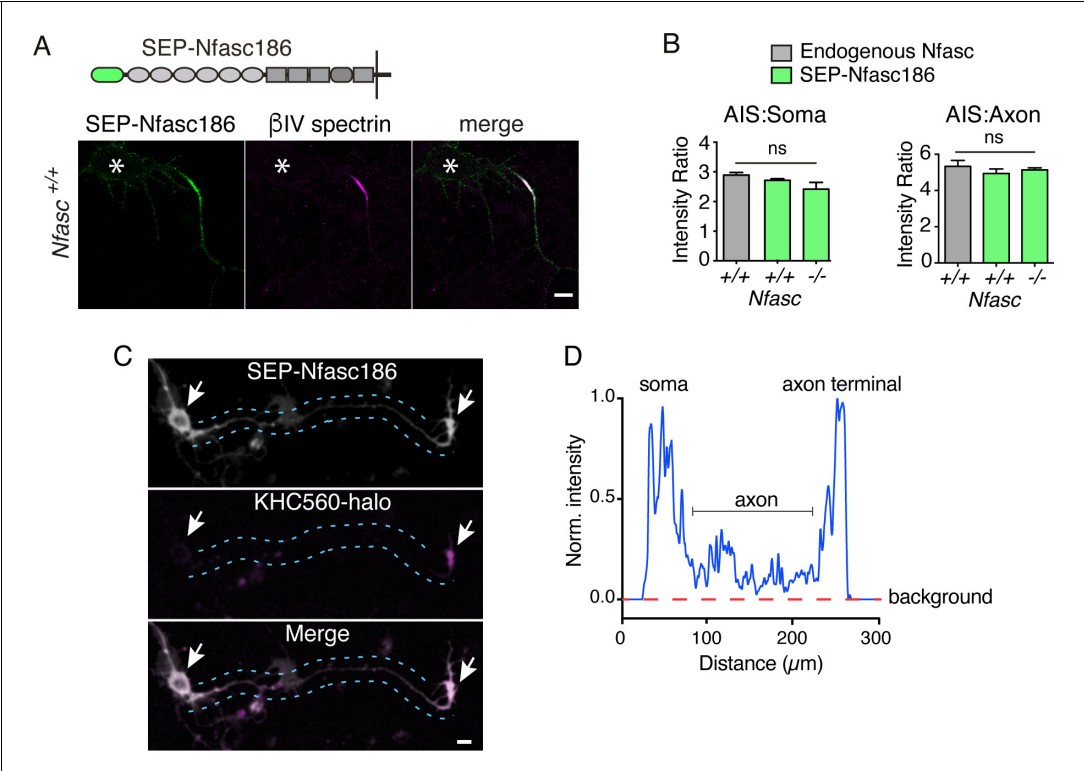

**Figure 1.** SEP-Nfasc186 accumulates at the AIS and the cell surface of the soma and axon terminus before the formation of the AIS. (**A**) Immunostaining of cortical neurons at DIV 7 shows that SEP-Nfasc186 is delivered to the AIS where it colocalises with ßIV-Spectrin. Location of the cell body is shown by asterisks. Scale bar, 10 µm. (**B**) Quantitation of signal intensity shows comparable enrichment of SEP-Nfasc186 relative to either the soma or distal axon when compared to endogenous Neurofascin irrespective of expression in WT or Neurofascin-null neurons. n = 3, ≥41 cells; one-way ANOVA; ns = not significant. (**C**) Live imaging before AIS formation at DIV 3 shows SEP-Nfasc186 at the surface of the soma and axon terminus (arrows). KHC560-halo identifies the axon terminus. Dashed lines outline the axon. Scale bar, 10 µm. (**D**) Line scan of top panel in (**C**) showing the SEP-Nfasc186 signal intensity in the cell body, axon and terminal relative to background.

The online version of this article includes the following figure supplement(s) for figure 1:

**Figure supplement 1.** SEP-Nfasc186 expressed in Neurofascin-null neurons and endogenous Neurofascin in WT cells accumulate at the cell surface of the soma and axon terminus before the formation of the AIS.

of the axon immediately proximal to the axon terminus to continual bleaching by FLIP (*Figure 3A*; *Video 2*). Imaging of a control region of the axon showed no diminution in overall fluorescence signal during the experiment (*Figure 3A–C*; *Video 2*). Loss of SEP-Nfasc186 signal proximal to the region of interest (ROI) indicated that SEP-Nfasc186 moves laterally in the axonal membrane from the axon terminus (*Figure 3A–C*; *Video 2*). Furthermore, since FLIP does not bleach vesicular SEP-Nfasc186, the loss of signal intensity at the axon terminal apparent in *Figure 3A* shows that diffusion of SEP-Nfasc186 in the axon membrane is bidirectional and can also occur anterogradely, as confirmed by asymmetric FLIP at the AIS with reference to *Figure 4* (see below).

Further evidence for the retrograde diffusion of Nfasc186 in the axonal membrane from the axon terminus came from photoconversion of Nfasc186-Dendra2. Photoconversion of Dendra2 from a green to a red state permits the tracking of protein movements in live cells (*Chudakov et al., 2007*). Photoconverted Nfasc186-Dendra2 in the axon terminal moved retrogradely in the distal axon (*Figure 3D*, and *Video 3*).

Video analysis of vesicles transporting Nfasc186-mCh shows their extensive anterograde and retrograde movement (*Video 4*). This is also evident for vesicles transporting Nfasc186-Dendra2 and kymographic analysis of their movement immediately proximal to the axon terminal showed that although nocodazole strongly inhibited vesicular transport of Nfasc186-Dendra2 (*Figure 3—figure supplement 1A and B*), it had no effect on the retrograde movement of photoconverted Nfasc186-Dendra2 (*Figure 3D and E*; *Video 3*). Hence, the signal arising from the retrograde movement of

photoconverted Nfasc186-Dendra2 visualised in the axon primarily reflects fluorescence from cell surface protein. In summary, Nfasc186 is extremely mobile after insertion in the neuronal membrane and can move towards the AIS in the plane of the axonal membrane.

To determine if delivery and retrograde diffusion from the axon terminal was unique to Nfasc186, we studied another AIS protein, the potassium channel Kv7.3. The fusion protein SEP-Kv7.3 shows a similar pattern of enrichment and delivery to the neuronal membrane at the cell body and axon terminus before the formation of the AIS (*Figure 3—figure supplement 2A–C*). Furthermore, FLIP at the axon immediately proximal to the axon terminal showed that Kv7.3 also undergoes retrograde movement from the axon terminus in the axonal membrane (*Figure 3—figure supplement 2D–F*).

Next, we wished to ask three questions: how mobile is SEP-Nfasc186 in the axonal membrane, is this mobility influenced by the axonal cytoskeleton and is the mobility of the protein changed at the AIS?

## Highly mobile Nfasc186 is delivered to the AIS by lateral diffusion in the axonal membrane

FRAP showed not only that SEP-Nfasc186 was highly mobile in the distal axonal membrane but also that its mobility was unaffected by either the inhibition of myosin II ATPase activity with Blebbistatin or disruption of microfilaments with latrunculin A (*Figure 4A and B*; *Berger et al., 2018*; *Sobotzik et al., 2009*). Hence, Nfasc186 can diffuse from the somatic or axon terminal plasma membrane to the AIS unassisted by the underlying cytoskeleton or its associated motor proteins (see also *Figure 3D*). The diffusion coefficient for Nfasc186 in the distal axon is $0.37 \pm 0.01$ μm$^2$/s and is comparable to the previously reported value for highly mobile, untethered axonal Nfasc186 ($0.34 \pm 0.02$ μm$^2$/s) (*Zhang et al., 2012*).

Nfasc186 stabilises the mature AIS (*Zonta et al., 2011*) but in order to monitor the trafficking of newly synthesised Nfasc186, when AIS assembly is at an early stage, we assessed Nfasc186 mobility at the AIS at DIV 5–6 and later at DIV 12–13, approximately 36 hr after transfection in each case (*Figure 4C and D*). Accumulation of SEP-Nfasc186 at the soma and axon terminal continued during AIS assembly (*Figure 4—figure supplement 1*). Maturation of the AIS was accompanied by a significant reduction in the mobility of SEP-Nfasc186 (recovery $46.1 \pm 0.8\%$ and $34.3 \pm 1.6\%$, respectively). In order to focus on the earlier stages of Nfasc186 recruitment, all subsequent studies on the AIS of cortical neurons were performed at DIV 3–6.

Since SEP-Nfasc186 was highly mobile in the axon membrane outside the AIS but much less

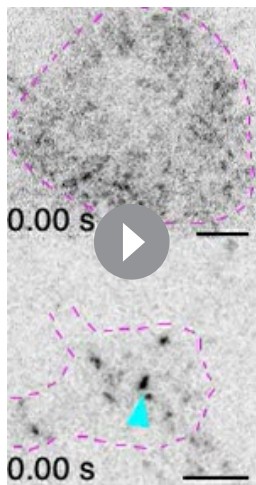

**Video 1.** TIRF microscopy imaging of SEP-Nfasc186 vesicle fusion events at the cell surface of the soma (top) and axon terminal (lower). The dashed lines outline the cell body and axon terminal respectively. Arrowheads point to some individual fusion events. Real interframe interval, 50 ms. Scale bar, 5 μm.
https://elifesciences.org/articles/60619#video1

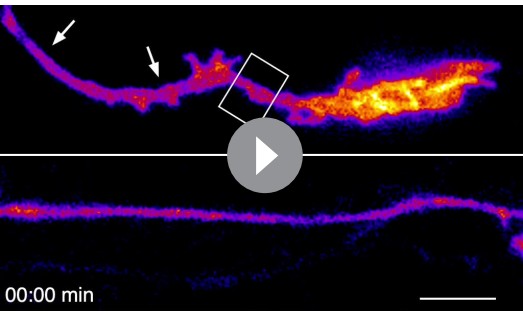

**Video 2.** FLIP of SEP-Nfasc186 in the distal axon proximal to the axon terminus. SEP-Nfasc186 fluorescence signal is depleted in the axon (arrows) proximal to the ROI (outlined by the box) following FLIP (top) and at the axon terminal itself showing that SEP-Nfasc186 moves retrogradely from and anterogradely to the axon terminus. The lower movie shows no significant bleaching of a control axon during the same acquisition period. Real interframe interval, 2 s. Scale bar, 5 μm.
https://elifesciences.org/articles/60619#video2

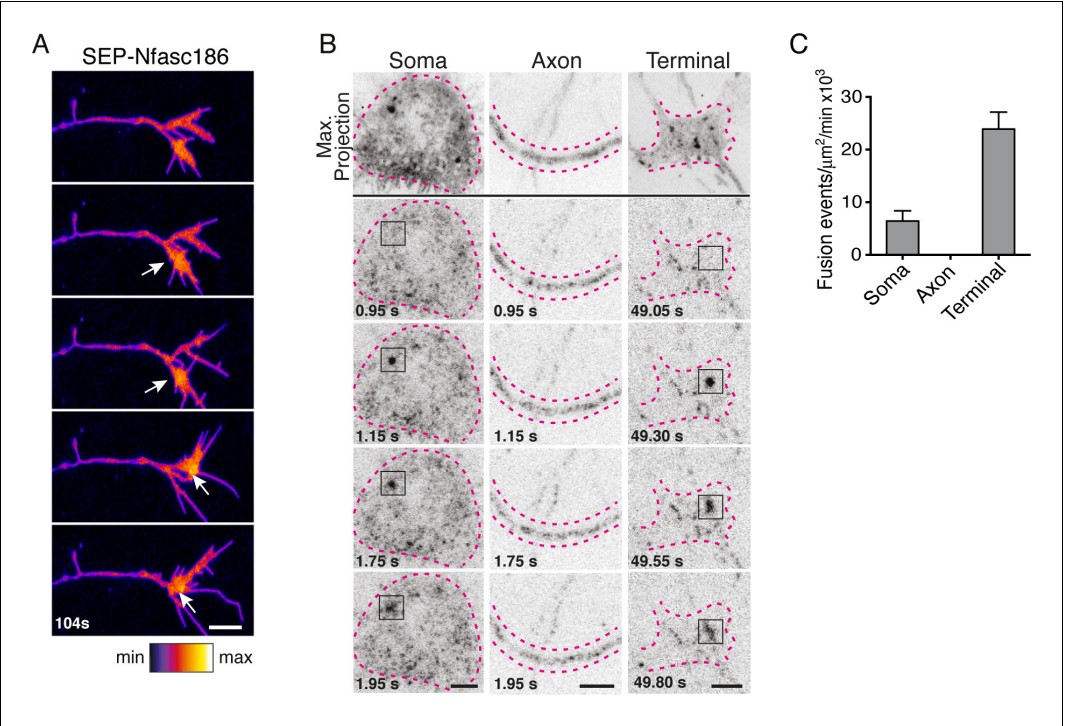

**Figure 2.** Nfasc186 is inserted into the neuronal membrane by vesicular fusion at the soma and axon terminus. (**A**) Still video images show transient elevated signal intensities (arrows) of SEP-Nfasc186 at the cell surface of an axon terminal. Scale bar, 5 μm. (**B**) TIRF microscopy reveals exocytotic insertion of SEP-Nfasc186 at the cell membrane of the soma and axon terminal (boxes), see *Video 1*. The soma, axon and terminal are outlined with dashed lines. Scale bar, 5 μm. (**C**) Quantitation of vesicle fusion events. Number of cells; soma = 4, axons = 7, terminal = 5.

mobile upon entry into the AIS, we wished to determine if the mobile pool contributed to the accumulation of Nfasc186 in the AIS. We combined FRAP with FLIP to determine the contribution by lateral diffusion of highly mobile protein to fluorescence recovery in the AIS since continual FLIP at regions flanking the FRAP ROI should selectively prevent fluorescence recovery by lateral ingress of fluorescent SEP-Nfasc186 at the AIS surface. FRAP-FLIP also permitted evaluation of the extent of direct fusion of axonal vesicles containing SEP-fusion proteins to fluorescence recovery at the AIS membrane surface since intra-axonal, vesicular SEP-Nfasc186, where the SEP fluorophore projects into the vesicular lumen, is neither fluorescent nor susceptible to continual photobleaching by FLIP: hence, any recovery in fluorescence must be due to vesicular fusion (*Figure 4—figure supplement 2A*; *Ashby et al., 2004*; *Ashby et al., 2006*; *Hildick et al., 2012*; *Makino and Malinow, 2009*; *Martin et al., 2008*; *Wilkinson et al., 2014*).

FRAP revealed substantial recovery of fluorescence within the AIS: however, this recovery is abolished by FLIP (*Figure 4E and F*; *Video 5*). We concluded that recovery of fluorescence is due to lateral movement of SEP-Nfasc186 in the axonal membrane with no significant contribution from direct vesicular fusion. By performing asymmetric FLIP on just one side of the ROI instead of bilaterally we were able to show that lateral diffusion of SEP-Nfasc186 into the AIS was

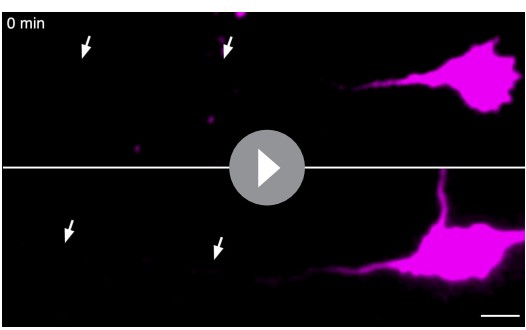

**Video 3.** Imaging of cortical neurons expressing Nfasc186-Dendra2 after photoconversion at the axon terminal. The photoconverted signal is propagated from the axon terminal to the distal axon (shown by arrows) in the absence (top) or presence of nocodazole (lower). Real interframe interval, 30 s. Scale bar, 5 μm.
https://elifesciences.org/articles/60619#video3

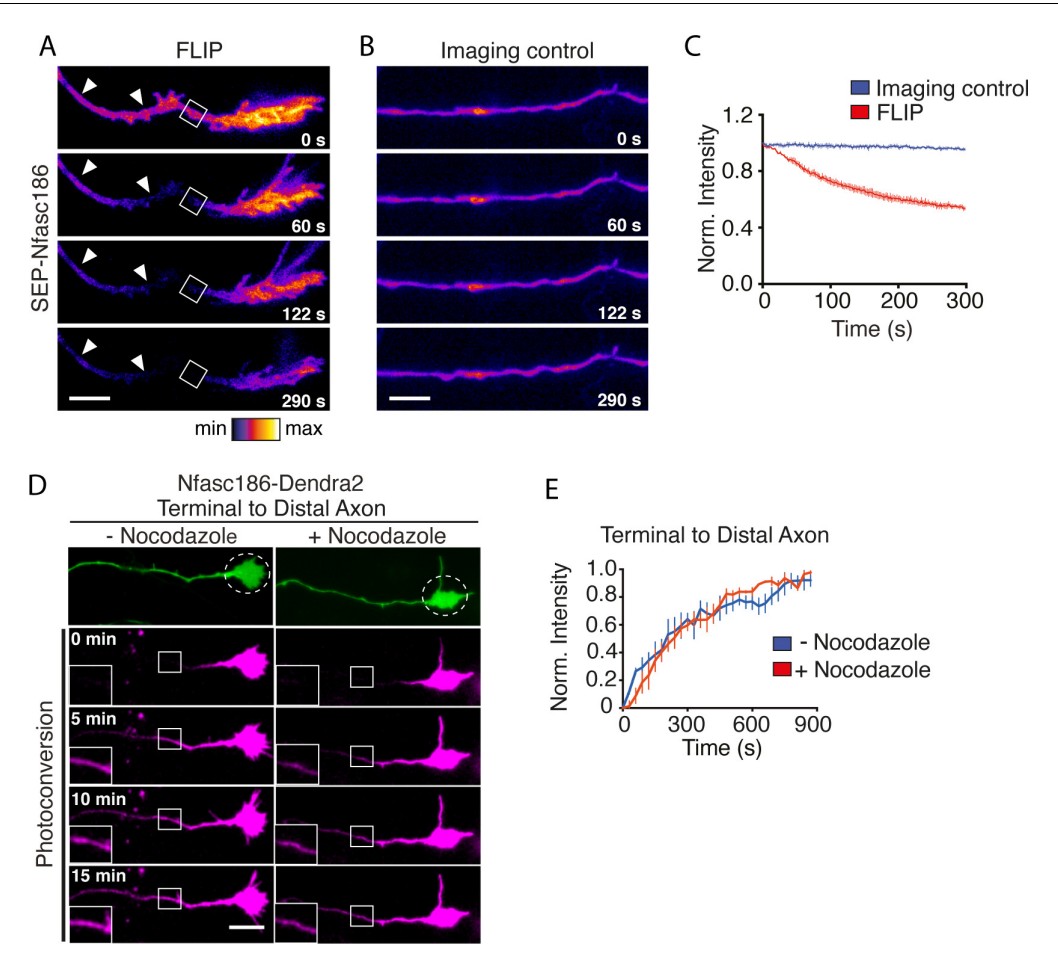

**Figure 3.** Lateral movement of Nfasc186 in the axon membrane from the axon terminal towards the distal axon. (A-C) Still images and quantitation from *Video 2* show depletion of SEP-Nfasc186 signal (arrowheads) proximal to the ROI after FLIP proximal to the axon terminal and at the axon terminal itself. In the imaging control axons were subjected to the same acquisition protocol without FLIP. n = 3, ≥15 cells. Scale bar, 10 μm. (D-E) Still images from *Video 3* of the photoconversion of Nfasc186-Dendra2 in the axon terminal and quantitation of normalised signal intensities (ROI boxes with insets) show that nocodazole does not affect Nfasc186-Dendra2 movement into the axon. An image in the green channel before photoconversion is shown in the top panel and the irradiated area is outlined in the dashed circle. n = 3, ≥14 cells. Scale bar, 10 μm.

The online version of this article includes the following figure supplement(s) for figure 3:

**Figure supplement 1.** Nocodazole inhibits the movement of vesicles transporting Nfasc186-Dendra2.
**Figure supplement 2.** Kv7.3 accumulates at the soma and axon terminus before the formation of the AIS.

bidirectional (recovery: 13.2 ± 0.2%-distal FLIP; 11.9 ± 0.7%-proximal FLIP; mean ± SEM, n = 3, Student's t test, not significant).

To confirm that the fate of SEP-Nfasc186 at the AIS was not influenced by the presence of excess endogenous Nfasc186 we also performed FRAP-FLIP on cortical neurons derived from Neurofascin-null mice and obtained similar results (*Figure 4—figure supplement 2B and C*). We conclude that fusion of Nfasc186 transport vesicles and concomitant protein insertion at the AIS itself is not a substantial source of surface-expressed AIS Nfasc186 in cortical neurons. In contrast, bidirectional lateral diffusion in the axonal membrane is the dominant mechanism by which Nfasc186 enters the AIS.

Discriminating between lateral diffusion and vesicular fusion as contributors to the recovery of fluorescence signal after FRAP-FLIP depends on the fact that not only is fluorescence emission from intra-axonal SEP-vesicular protein eclipsed, but also that this population is not susceptible to bleaching or photochemical damage. Several previous studies have shown these assumptions to be correct

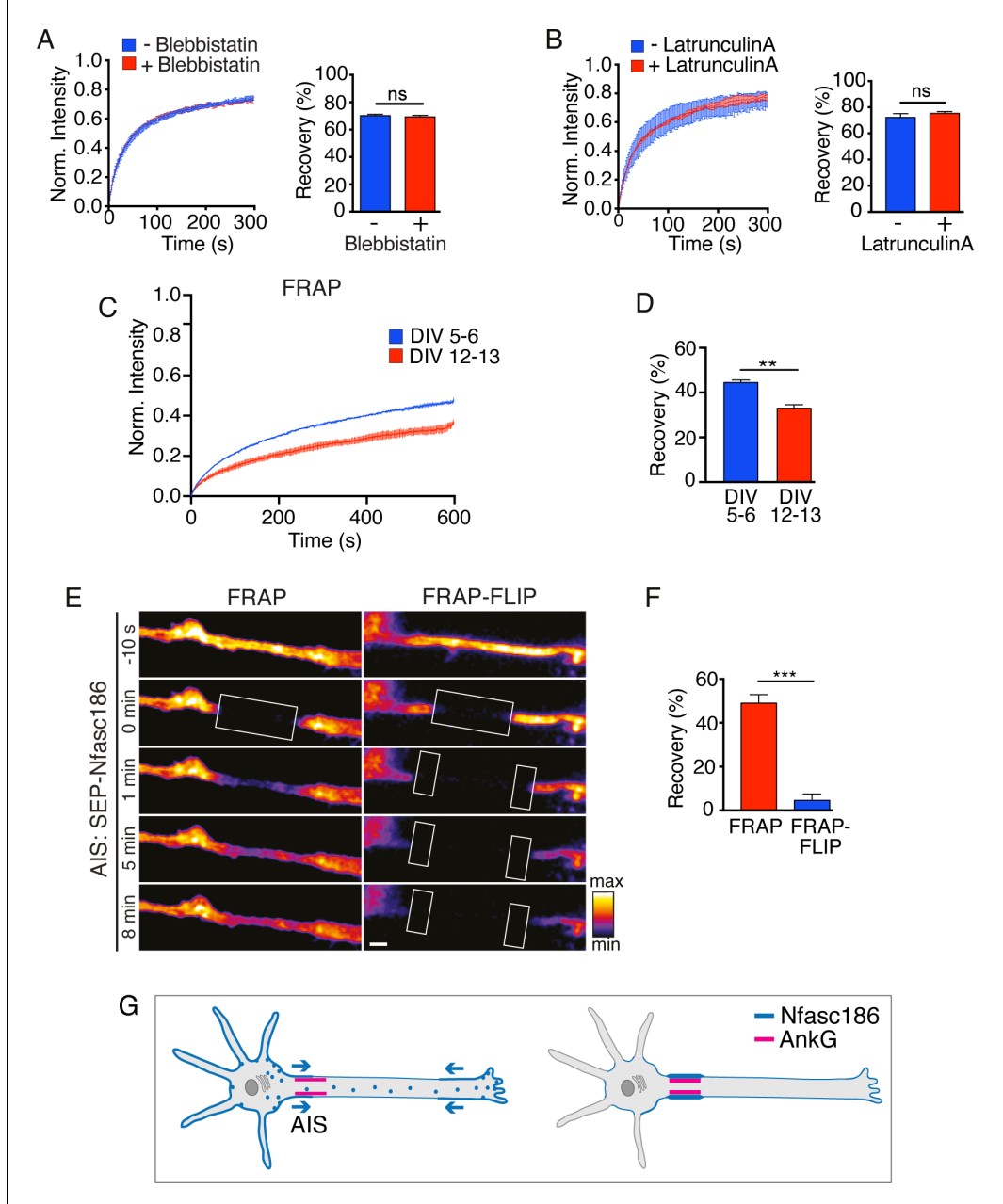

**Figure 4.** Nfasc186 is delivered to the AIS by lateral diffusion in the cell membrane of cortical neurons. (**A—B**) Cultured cortical neurons were treated with the myosin ATPase inhibitor Blebbistatin and latrunculin A. The FRAP curves show that the drugs did not affect recovery of the mean signal intensity from three independent experiments for each condition. The bar graph shows the mean recovery fraction. n = 3, ≥16 cells; Student's t test; ns = not significant. (**C-D**) Comparison of FRAP curves at DIV 5–6 and DIV 12–13 shows that SEP-Nfasc186 becomes significantly more immobilised at the AIS with time. n = 3, ≥17 cells. Student's t test. **p<0.01. (**E-F**). Still images from *Video 5* of FRAP and FRAP-FLIP within the AIS (FRAP at boxed ROIs and FLIP at flanking boxed ROIs) and quantitation show that signal recovery after photobleaching is prevented by FLIP. n = 3, ≥16 cells; Student's t test; ***p<0.001. Scale bar, 2 μm. (**G**) Model depicting bidirectional delivery of Nfasc186 to the AIS.

The online version of this article includes the following figure supplement(s) for figure 4:

**Figure supplement 1.** Enrichment of SEP-Nfasc186 at the soma and axon terminal during AIS assembly.

**Figure supplement 2.** SEP-Nfasc186 is delivered to the AIS by lateral diffusion in Neurofascin-null cortical neurons.

**Figure supplement 3.** Control experiment to show that the vesicular fraction of SEP-Nfasc186 is not bleached during FRAP-FLIP.

**Figure supplement 4.** Nfasc186 is recruited to the AIS by lateral diffusion in cerebellar Purkinje Neurons.

**Figure supplement 5.** AnkG immobilises Nfasc186 at the AIS by its interaction with AnkG but delivery to the axon terminal surface is independent of AnkG.

**Video 4.** Transport of Nfasc186-mCh in axonal vesicles in rat cortical axons. Arrows indicate the directionality of movement. Kymograph analysis of 5 axons showed that the anterograde velocity was 2.3 ± 0.27 μm/s and retrograde velocity was 1.9 ± 0.18 μm/s (mean ± SEM). Real interframe interval, 0.5 s. Scale bar, 5 μm.
https://elifesciences.org/articles/60619#video4

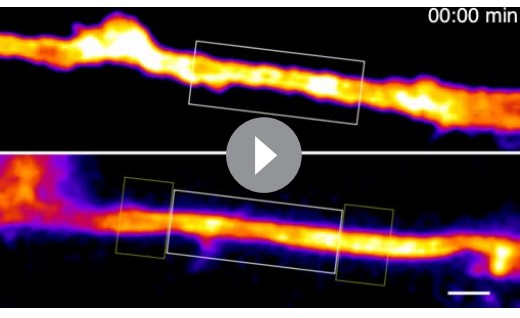

**Video 5.** FRAP (top) and FRAP-FLIP (lower) at the AIS of cortical neurons expressing SEP-Nfasc186 (DIV 5). The boxes indicate the FRAP ROI and the flanking FLIP ROIs. Real interframe interval, 2 s. Scale bar, 2 μm.
https://elifesciences.org/articles/60619#video5

(*Ashby et al., 2004*; *Ashby et al., 2006*; *Hildick et al., 2012*; *Makino and Malinow, 2009*; *Martin et al., 2008*; *Wilkinson et al., 2014*). Nevertheless, we wished to confirm that SEP does indeed report Nfasc186 exclusively at the neuronal surface (*Figure 4—figure supplement 3A*), and, using a refined protocol, we confirmed that intra-axonal vesicular SEP-Nfasc186 is neither fluorescent nor susceptible to photobleaching (*Figure 4—figure supplement 3B–F*).

In order to extend the conclusions from these data to other neuronal cell types, we asked if Nfasc186 is recruited to the AIS of neurons in an organotypic preparation by lateral diffusion. We have previously established that Nfasc186 has an essential role in stabilising the AIS of Purkinje cells in vivo (*Zonta et al., 2011*). FRAP-FLIP analysis of acute cerebellar slices from P10 transgenic mice showed that fluorescence recovery at the Purkinje cell AIS was indeed by lateral diffusion in the plane of the axonal membrane, as found for cortical neurons (*Figure 4—figure supplement 4A–D*).

## AnkG immobilises Nfasc186 at the AIS but is not required for Nfasc186 delivery to the axonal membrane

AnkG is believed to act as a pioneer constituent and key organizer of the nascent AIS (*Dzhashiashvili et al., 2007*; *Galiano et al., 2012*; *Hedstrom et al., 2008*; *Jenkins and Bennett, 2001*). Numerous studies have emphasized the importance of the interaction of Nfasc186 with AnkG at the AIS and we confirmed that mutation of the AnkG binding site prevents SEP-Nfasc186YA accumulation at the AIS (*Boiko et al., 2007*; *Davis and Bennett, 1994*; *Fréal et al., 2019*; *Lemaillet et al., 2003*; *Zhang et al., 1998*; *Zonta et al., 2011*; *Figure 4—figure supplement 5A–C*). FRAP confirmed that the mutant protein was indeed highly mobile at the AIS of cortical neurons (*Figure 4—figure supplement 5D–F*). Nevertheless, the accumulation and insertion of Nfasc186 at the surface membrane of the axon terminal does not require interaction with AnkG (*Figure 4—figure supplement 5G*). Hence, neither the insertion of Nfasc186 into the neuronal membrane nor its mobility in the axonal membrane requires the cotransport of an AnkG/AIS membrane protein complex. Further, although the fractional recovery of SEP-Nfasc186 at the AIS declined between 5–6 and 12–13 days in culture (*Figure 4D*), the AnkG binding mutant of Nfasc186 retained high mobility (recovery 78.6 ± 2.0% and 79.1 ± 1.5%, respectively) during the same period (*Figure 4—figure supplement 5E–F*). This shows that as Nfasc186 becomes increasingly immobilised during the early stages of AIS formation (*Figure 4C*), it is the interaction of Nfasc186 with AnkG that is overwhelmingly important in anchoring and immobilizing Neurofascin at the surface of the AIS.

The routes by which Nfasc186 is recruited to the AIS are depicted in a model shown in *Figure 4G*. Our model indicates that vesicles that transport Nfasc186 are able to fuse not only with the somatic plasma membrane but also distally to the axon terminal membrane, but they do not insert Nfasc186 at the AIS directly. Since Nfasc186 has a major role in assembling the node of Ranvier in myelinated axons, this model of sequential membrane delivery and clustering may also inform studies on how transmembrane proteins are recruited to the node (*Davis et al., 1996*; *Sherman et al., 2005*; *Tait et al., 2000*; *Zhang et al., 2012*; *Zonta et al., 2008*).

# Materials and methods

**Key resources table**

| Reagent type (species) or resource | Designation | Source or reference | Identifiers | Additional information |
|---|---|---|---|---|
| Strain, strain background (*R. norvegicus*, male and female) | Sprague-Dawley Crl: CD(SD) | Charles River Laboratories | RRID:RGD_734476 | University of Edinburgh maintained colony |
| Strain, strain background (*M. musculus*, male and female) | *Nfasc*$^{-/-}$ mice Background: C57BL/6JOla | *Sherman et al., 2005* | | Peter Brophy, University of Edinburgh |
| Strain, strain background (*M. musculus*, male and female) | L7-SEP-Nfasc186 Background: C57BL/6JOla | This paper | | Peter Brophy, University of Edinburgh |
| Transfected construct (*M. musculus*) | SEP-Nfasc186-pCMV5a | This paper | | Peter Brophy, University of Edinburgh |
| Transfected construct (*M. musculus*) | SEP-Nfasc186YA-pCMV5a | This paper | | Peter Brophy, University of Edinburgh |
| Transfected construct (*M. musculus*) | Nfasc186-mCh-pCMV5a | This paper | | Peter Brophy, University of Edinburgh |
| Transfected construct (*M. musculus*) | Nfasc186-Dendra2-pCMV5a | This paper | | Peter Brophy, University of Edinburgh |
| Transfected construct (human) | SEP-Kv7.3-pCDNA3.1 | *Benned-Jensen et al., 2016* | | Nicole Schmitt, University of Copenhagen |
| Transfected construct (human) | Kv-7.2-pXOOM | *Benned-Jensen et al., 2016* | | Nicole Schmitt, University of Copenhagen |
| Transfected construct (*R. norvegicus*) | AnkG-mCh | Addgene *Leterrier et al., 2011* | plasmid #42566 | |
| Transfected construct (human) | KHC560-halo | *Twelvetrees et al., 2016* | | Alison Twelvetrees, University of Sheffield |
| Antibody | Neurofascin (rabbit polyclonal) | *Tait et al., 2000* | | Intracellular epitope IF (1:1000) |
| Antibody | Neurofascin (mouse monoclonal) | UC Davis/NIH NeuroMab | clone: A12/18 | Extracellular epitope IF (1:10) |
| Antibody | ßIV spectrin (rabbit polyclonal) | *Zonta et al., 2011* | | IF (1:200) |
| Antibody | GFP (chicken polyclonal) | Abcam | Cat# ab13970 | IF (1:1000) |
| Antibody | Ankyrin G (mouse monoclonal) | UC Davis/NIH NeuroMab | clone: N106/65 | IF (1:30) |
| Antibody | Anti-Rabbit Alexa Fluor 594 | Jackson ImmunoResearch | Cat# 111-585-14 | IF (1:1000) |
| Antibody | Anti-Chicken Alexa Fluor 488 | Jackson ImmunoResearch | Cat# 703-545-155 | IF (1:1000) |
| Antibody | Anti-Mouse IgG2a Alexa Fluor 488 | Invitrogen | Cat# A-21131 | IF (1:1000) |
| Antibody | Anti-Mouse IgG2b Alexa Fluor 568 | Invitrogen | Cat# A-21144 | IF (1:1000) |
| Chemical compound, drug | Phusion High-Fidelity DNA Polymerase | New England BioLabs | Cat# M0530S | |

*Continued on next page*

*Continued*

| Reagent type (species) or resource | Designation | Source or reference | Identifiers | Additional information |
|---|---|---|---|---|
| Chemical compound, drug | T4 DNA Ligase | Thermo Fisher Scientific | Cat# EL0011 | |
| Chemical compound, drug | DpnI | New England BioLabs | Cat# R0176S | |
| Chemical compound, drug | Lipofectamine 2000 Transfection Reagent | Thermo Fisher Scientific | Cat#11668030 | |
| Chemical compound, drug | DMSO | Sigma-Aldrich | Cat# 434302 | |
| Chemical compound, drug | Poly-D-lysine | Sigma-Aldrich | Cat# P6407 | |
| Chemical compound, drug | B-27 | Thermo Fisher Scientific | Cat# 17504044 | |
| Chemical compound, drug | Fish skin gelatin | Sigma-Aldrich | Cat# G7765 | |
| Chemical compound, drug | Nocodazole | Sigma-Aldrich | Cat# SML1665 | |
| Chemical compound, drug | Latrunculin A | Merck | Cat# 428026 | |
| Chemical compound, drug | (S)-nitro-Blebbistatin | Cayman Chemical | Cat# 85692575–2 | |
| Chemical compound, drug | JF549-Halo Tag Ligand | Janelia Research Campus *Grimm et al., 2017* | | |
| Sequence-based reagent | Mutagenesis primer one to insert AgeI site in Nfasc cDNA | Integrated DNA Technologies | This paper | GAATGAGCTGACCGGTC AACCCCCAACTATCAC |
| Sequence-based reagent | Mutagenesis primer two to insert AgeI site in Nfasc cDNA | Integrated DNA Technologies | This paper | GGGGGTTGACCGGTCAG CTCATTCTGAATGCTTG |
| Sequence-based reagent | Mutagenesis primer one to generate Nfasc186YA | Integrated DNA Technologies | This paper | AAGGAGCCATCTTCATTG |
| Sequence-based reagent | Mutagenesis primer two to generate Nfasc186YA | Integrated DNA Technologies | This paper | TATTGGCCAGGCCACTGTCAAAAAG |
| Sequence-based reagent | Dendra2-HindIII-fwd | Integrated DNA Technologies | This paper | AAAAAGCTTGGAGGAACCATGAAC ACCCCGGGAATTAACC |
| Sequence-based reagent | Dendra2-SalI-rev | Integrated DNA Technologies | This paper | TTTGTCGAC TCACCACACCTGGCTGGGCA |
| Software, algorithm | FIJI | *Schindelin et al., 2012* | RRID:SCR_002285 | https://imagej.net/Fiji |
| Software, algorithm | Prism 6.0 | GraphPad | RRID:SCR_002798 | |
| Software, algorithm | KymoTool Box | *Zala et al., 2013* | | Frédéric Saudou, University of Grenoble Alpes |

## Animals

Animal work was performed according to UK legislation (Scientific Procedures) Act 1986 and the guidelines of the University of Edinburgh Ethical Review policy. The generation of *Nfasc*<sup>-/-</sup> mice has been described (*Sherman et al., 2005*). To generate SEP-Nfasc186 transgenic mice a SEP-Nfasc186 transgene was constructed by inserting a restriction site (Age I) by site-directed mutagenesis in the murine Nfasc186 cDNA (*Zonta et al., 2008*) at amino acid 38 between the signal sequence and the first IgG domain. Super-ecliptic pHluorin (SEP) cDNA (a gift from Dr. Gero Miesenböck, University of Oxford) was cloned into the Age I site and then inserted into a plasmid containing the cerebellar Purkinje cell-specific L7 promoter (*Oberdick et al., 1990*). Transgenic mice were generated by

pronuclear injection as described (*Sherman and Brophy, 2000*). All mice were backcrossed to a C57BL/6 background for at least 10 generations.

## Cortical neuron culture

Primary cortical neurons were prepared from postnatal day P0-P1 Sprague-Dawley rats irrespective of sex. Cortices were isolated and meninges were removed; the tissue was dissociated using an enzymatic solution of papain (45 U/ml; Worthington Biochemical Corp.), L-cysteine (0.2 mg/ml; Sigma-Aldrich) and DNase I (0.40 mg/ml; Sigma-Aldrich) for 15 min at 37 °C. The reaction was stopped by adding Ovomucoid protease inhibitor (1 µg/ml; Worthington Biochemical Corp.). Thereafter, neurons were dissociated in seeding media containing DMEM (Gibco, Life Technologies) supplemented with 10% fetal bovine serum (FBS, Gibco, Life Technologies), 1% GlutaMAX (Gibco, Life Technologies) and 1% penicillin/streptomycin (Sigma- Aldrich). Prior to dissection, 35 mm glass-bottom dishes (ibidi, MatTek) and 13 mm glass coverslips (VWR) were coated with poly-D-lysine (100 µg/ml; Sigma-Aldrich) overnight. Neurons were seeded at a density 60,000 cells/100 µl in culture medium. After 2 hr, the medium was changed to neurobasal medium (Gibco, Life Technologies), supplemented with 2% B-27, 1% GlutaMAX and 1% penicillin/streptomycin. 5-fluoro-2'-deoxyuridine (10 µM, Sigma-Aldrich) was added to cultured neurons to inhibit the growth of non-neuronal cells. The cultures were incubated at 37°C in a humidified atmosphere containing 5% $CO_2$.

## Organotypic cerebellar slice culture

Brains from L7-SEP-Nfasc186 transgenic mice at postnatal day P9–P10 were placed in ice-cold Hank's Balanced Salt Solution (HBSS; Gibco), supplemented with glucose (5 mg/ml; Gibco) and 1% penicillin/streptomycin. The meninges and forebrain were immediately removed. Parasagittal cerebellar slices (100 µm) were cut using a vibratome (Leica VT-1000S) and placed in culture medium composed of 50% MEM (Gibco), 25% HBSS, 25% heat-inactivated horse serum (Sigma-Aldrich), glucose (5 mg/ml), 1% GlutaMAX and 1% penicillin/streptomycin. The slices were transferred to the membrane of 30 mm cell culture insert (Millicell, Millipore) on prewarmed medium and were maintained at 37°C in a humidified atmosphere containing 5% $CO_2$. Live imaging was performed after 3–4 hr in Hibernate-A-Low Fluorescence medium (BrainBits) supplemented with 2% B27% and 1% GlutaMAX (Hibernate-A imaging medium).

## DNA constructs and transfection

SEP-Nfasc186 was subcloned into the mammalian expression vector pCMV5a. The ankyrin G binding mutant of Nfasc186, SEP-Nfasc186YA, was generated by site-directed mutagenesis of the conserved FIGQY domain to FIGQA (*Boiko et al., 2007*; *Zhang and Bennett, 1998*). To generate the Nfasc186-mCh construct, mCherry (mCh) was fused to the C-terminus of mouse Nfasc186 cDNA and subcloned into pCMV5a. Dendra2 (Evrogen, [*Gurskaya et al., 2006*]) was fused to the C-terminus of the full-length Nfasc186 and cloned into the pCMV5a vector. The following plasmids were gifts: AnkG-mCh (*Leterrier et al., 2011*), SEP-Kv7.3, Kv7.2 (*Benned-Jensen et al., 2016*) KHC560-halo (*Twelvetrees et al., 2016*). The constructs were expressed by transient transfection using Lipofectamine 2000 Transfection Reagent (Life Technologies).

## Live cell imaging

Live imaging was performed using an inverted wide-field microscope (Zeiss Axio Observer), equipped with the following objectives: Plan Apochromat 20X (NA 0.8; Zeiss), Plan 40X oil (NA 1.3; Zeiss), Plan Apochromat 63X oil (NA 1.4; Zeiss), Alpha Plan Apochromat 100X oil (NA 1.46; Zeiss), together with Definite Focus.2 (for Z-drift correction), an ORCA-Flash4.0 V2 Digital CMOS camera (Hamamatsu Photonics) and a 37 °C imaging chamber (PeCon) in a humidified atmosphere containing 5% $CO_2$. LED illumination (Colibri 7, Zeiss) was used for image acquisition and camera pixel size was binned to 2 × 2 to achieve better signal-to-noise ratios. The entire imaging workflow was controlled by Zeiss imaging software (ZEN 2.3 blue edition). In order to perform photomanipulation, the microscope was coupled to two diode lasers (473 nm and 405 nm) and a laser scanning device (UGA-42 Firefly, Rapp OptoElectronic). Lasers were controlled using SysCon software, synchronised to image acquisition by ZEN 2.3. For experiments utilising SEP-Nfasc186, SEP-Nfasc186YA and SEP-Kv7.3 the medium was replaced with SEP imaging medium (140 mM NaCl, 5 mM KCl, 15 mM

D-glucose, 1.5 mM CaCl$_2$, 1.5 mM MgCl$_2$, 20 mM HEPES, pH 7.4). An acidic SEP imaging medium was used to quench surface fluorescence in which MEM replaced HEPES and the pH adjusted to 6.0. To allow subsequent recovery of SEP fluorescence the medium was changed to 50 mM NH4Cl, 90 mM NaCl, 5 mM KCl, 15 mM glucose, 1.8 mM CaCl$_2$, 0.8 mM MgCl$_2$, 20 mM HEPES, pH 7.4. The culture medium for experiments utilising Nfasc186-Dendra2 was Hibernate-A-Low Fluorescence medium (BrainBits) supplemented with 2% B27% and 1% GlutaMAX (Hibernate-A imaging medium).

Kinesin560-halo (KHC560-halo) was expressed in neurons at DIV 2–3 either with SEP-Nfasc186 or in combination with SEP-Kv.7.3 and Kv7.2. The axon terminal was identified by expression of KHC560-halo visualised by incubating the neurons with the halo-ligand conjugated to Janelia Fluor-549 fluorophore (100 nM) (JF549-HaloTag Ligand) (*Grimm et al., 2017*) for 10 min at 37°C followed by washes with SEP imaging medium. Unless specified otherwise, transfected neurons were imaged using low LED power (10%) for 5 min at 1 s intervals with a 100 ms exposure time.

## Fluorescence recovery after photobleaching

For fluorescence recovery after photobleaching (FRAP) experiments, cortical neurons were cotransfected at DIV 3–4 or DIV 10–11 with cDNAs encoding AnkG-mCh with either SEP-Nfasc186 or SEP-Nfasc186YA. The AIS was identified by AnkG-mCh expression after ~36 hr. Imaging was performed at 37 °C using the SEP imaging medium. A region of the AIS was photobleached using a 473 nm laser (50% for ~500 ms). Pre-bleach and post-bleach frames were acquired at the rate of 1 frame every 2 s for 10 s and 10 min, respectively. Axonal FRAP experiments were conducted using the same experimental parameters except post-bleaching acquisition was for 5 min.

## Fluorescence loss in photobleaching (FLIP)

Neurons were transfected either with SEP-Nfasc186 or in combination with SEP-Kv.7.3 and Kv7.2. An area proximal to the axon terminal was repeatedly photobleached as described below and imaged at intervals of 2 s for 5 min. A low laser power setting (15%,100 ms) was used to avoid phototoxicity.

## FRAP-FLIP

The FRAP-FLIP protocol was adapted from the method previously described by Henley and colleagues (*Hildick et al., 2012*). SEP-Nfasc186 and AnkG-mCh were co-expressed in cortical neurons by transfection at DIV 3–4 and experiments were performed at DIV 5–6. For FRAP, a single region of interest (ROI) within the AIS was photobleached as described above and allowed to recover for 10 min; During this period of acquisition two flanking ROIs were repeatedly photobleached using the 473 nm laser (15% for ~100 ms) to achieve effective photobleaching and imaged at intervals of 2 s. Laser power settings for the FRAP-FLIP experiments were carefully evaluated to ensure neuron viability was not compromised as evaluated by the recovery of the fluorescent signal at the FLIP-ROI 15 min after the end of the experiment. Axonal FRAP-FLIP experiments were conducted using the same experimental parameters except post-bleaching acquisition was for 5 min.

## Photoconversion

Neurons were transfected with Nfasc186-Dendra2 cDNA at DIV 3–4 and live cell imaging was performed in Hibernate-A imaging medium 16–20 hr after transfection. A 40X objective was used to identify transfected neurons (green fluorescence). Photoconversion was performed either at the soma or axon terminal using a low laser power with a wavelength of 405 nm (1–2%), with 5–6 exposures, each with a duration of ~700 ms. Once the selected area was converted, the axon was imaged using the 63X objective, a multiband pass filter (Chroma Technology Corp) and LED illumination. Images were acquired every 30 s for 15 min (terminal) or 30 min (soma). To assess the consequence of disrupting microtubules, photoconversion was performed after incubation with 20 µM nocodazole in DMSO for 1 hr, and control cells received DMSO alone.

## Vesicle trafficking

For studies on vesicle tracking, neurons at DIV 3–4 were transfected with either Nfasc186-mCh or Nfasc186-Dendra2 cDNA. After 16–20 hr of transfection, live cell imaging was performed in Hibernate-A imaging medium. Images were recorded every 500 ms with the 100X (NA 1.46) objective and

an exposure time of 100 ms. Vesicle movement was analysed using kymographs generated by an ImageJ plugin KymoToolBox (*Zala et al., 2013*). The kymographs were manually traced to obtain vesicle speed.

## TIRF microscopy

Neurons were cultured on 35 mm glass-bottom dishes (170 ± 5 μm thickness, ibidi) and cotransfected with SEP-Nfasc186 and KHC560-halo cDNAs at DIV 3. Imaging was performed 16–18 hr after transfection. To visualise axon terminals JF 549 Halo Tag Ligand was first added to the neurons for 10 min. SEP imaging medium was added to the cultures after washing. TIRF experiments were conducted using an inverted Zeiss TIRF III microscope with a 488 nm laser, a 100X Alpha Plan Apochromat oil immersion objective (NA 1.46, Zeiss) and TIRF III motorised slider in a closed environmental chamber at 37 ˚C. The illumination angle was set for evanescent illumination (~110 nm) (*Axelrod, 2001*). Images were acquired with a Photometrics Evolve Delta EMCCD camera every 50 ms for 1—2 min, using Zen Blue 2.3 software.

## Drug treatments

The myosin II ATPase inhibitor Blebbistatin (20 μM, (S)-nitro-Blebbistatin) was added to the neuronal cultures for ~20 hr before FRAP experiments. Cortical neurons were treated with latrunculin A (5 μM) for 1 hr before FRAP experiments. Nocodazole treatment to disrupt microtubules was as described above.

## Immunofluorescence

Cultured cortical neurons were fixed by immersion in 4% paraformaldehyde (PFA) in 0.1 M sodium phosphate buffer (pH 7.4) for 15 min at room temperature, followed by three washes in PBS. Brains from WT and L7-SEP-Nfasc186 mice at P10 were fixed by transcardial perfusion with 4% PFA in 0.1 M sodium phosphate buffer (pH 7.4) as described previously (*Tait et al., 2000*). Brains were postfixed for 30 min with 4% PFA in 0.1 M sodium phosphate buffer, followed by three washes in PBS. Parasagittal vibratome sections (100 μm) were cut. Fixed samples were blocked (cortical neurons for 30 min, cerebellar slices for 1 hr) in blocking buffer containing 5% fish skin gelatin, and Triton X-100 (cortical neurons 0.2%, cerebellar slices 0.5%) in PBS followed by incubation with primary antibodies for 2 hr or overnight. Primary antibodies were diluted in 5% fish skin gelatin for cortical neurons and in blocking buffer for cerebellar slices. Primary antibodies used in the study are: GFP, ßIV spectrin (*Zonta et al., 2011*), AnkyrinG and Neurofascin (intracellular [*Tait et al., 2000*]). For surface labelling of Neurofascin, live cells were incubated with anti-Neurofascin (extracellular) antibody (diluted in the neurobasal culture media) for 30 min at 37 ˚C followed by fixation and further staining with Alexa Fluor 568-conjugated phalloidin (1:200, Invitrogen) and secondary antibodies. The Alexa Fluor conjugated secondary antibodies were diluted in 5% fish skin gelatin for cortical neurons and in blocking buffer for cerebellar slices and were incubated for 2 hr. Samples were mounted in Vectashield Mounting Medium (Vector Laboratories). For AIS intensity analysis, cortical neuron images were acquired on a Zeiss Axio Observer with a 63X objective lens. Representative images were acquired on a Zeiss LSM710 confocal microscope with a Plan Apochromat 63X oil objective (NA 1.4; Zeiss). Images from cerebellar slices were acquired on a Leica TCL-SL confocal microscope equipped with a 63X objective lens (NA 1.4) using Leica proprietary software.

## Quantification and statistical analysis

FIJI was used to view and analyse images and videos. The intensity profile and total signal intensity of the AIS and distal axons were measured in FIJI. For FRAP and FRAP-FLIP analysis, the mean fluorescence intensity of the bleached region was normalised to the intensity of the pre-bleached region and plotted as a fraction after background correction of all frames. The normalised data were fitted with a single-exponential equation to extract the recovery fraction after photobleaching. The diffusion coefficient was estimated by fitting the recovery data to a one-dimensional diffusion model (*Ellenberg and Lippincott-Schwartz, 1999*). For FLIP analysis at the axon terminus, an ROI was selected proximal to the bleaching region; the average signal intensity of each frame was measured and plotted as a fraction of the initial signal intensity before imaging.

In order to quantify Nfasc186-Dendra2 movement to the axon, an ROI was selected in the axon. The ROI was at a constant distance from the axon terminal to allow comparison between different experiments. The average signal intensity of the ROI in each frame was measured and plotted as a fraction of the peak signal intensity.

All data are represented as mean ± SEM unless otherwise mentioned in the figure legends. Statistical analyses were performed using GraphPad Prism version 6.0 software. Statistical significance was analysed by two-tailed Student's t-test or one-way ANOVA followed by Tukey's multiple comparisons test. n values are reported in the corresponding figure legends. The sample size was determined based on similar studies within the field. A p-value$<0.05$ was considered statistically significant.

## Acknowledgements

We thank Qiushi Li for invaluable technical support. This work was supported by a grant from the Wellcome Trust to PJB (Grant No. 107008/Z/15/Z). PJB is a Wellcome Trust Investigator.

## Additional information

### Funding

| Funder | Grant reference number | Author |
| --- | --- | --- |
| Wellcome | 107008 | Aniket Ghosh<br>Elise LV Malavasi<br>Diane L Sherman<br>Peter J Brophy |

The funders had no role in study design, data collection and interpretation, or the decision to submit the work for publication.

### Author contributions

Aniket Ghosh, Conceptualization, Formal analysis, Validation, Investigation, Methodology, Writing - original draft, Writing - review and editing; Elise LV Malavasi, Methodology, Writing - review and editing; Diane L Sherman, Writing - original draft, Project administration, Writing - review and editing; Peter J Brophy, Conceptualization, Data curation, Formal analysis, Supervision, Funding acquisition, Validation, Investigation, Writing - original draft, Project administration, Writing - review and editing

### Author ORCIDs

Aniket Ghosh  https://orcid.org/0000-0002-3771-6390
Diane L Sherman  https://orcid.org/0000-0002-3104-6656
Peter J Brophy  https://orcid.org/0000-0002-0262-9545

### Ethics

Animal experimentation: All animal work was performed according to UK legislation (Scientific Procedures) Act 1986 according to the guidelines of and approved by the University of Edinburgh Animal Welfare and Ethical Review Body. All work was performed under a Project Licence (No. P0F4A25E9) issued by the UK Home Office to Peter Brophy and this licence is in force until 26 March, 2022.

### Decision letter and Author response

Decision letter https://doi.org/10.7554/eLife.60619.sa1
Author response https://doi.org/10.7554/eLife.60619.sa2

## Additional files

### Supplementary files
• Transparent reporting form

### Data availability
All data generated or analysed during this study are included in the manuscript and supporting files.

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
