## [Decision Letter]

**Acceptance summary:**

The study provides an interesting mechanism to account for the insertion of neurofascin 186 and the Kv7.3 channel to the axon initial segment. The delivery and fate of these proteins is supported by well performed experiments. Several suggestions made to strengthen the manuscript are addressed in the revision. The quantitation of Kv7.3 and neurofascin has been addressed with clarification and new videos that follow the movement of the two proteins.

**Decision letter after peer review:**

Thank you for submitting your article "The key axon initial segment protein Neurofascin is delivered to the neuronal membrane at the soma and axon terminal" for consideration by *eLife*. Your article has been reviewed by three peer reviewers, and the evaluation has been overseen by a Reviewing Editor and Kenton Swartz as the Senior Editor. The following individual involved in review of your submission has agreed to reveal their identity: Vann Bennett (Reviewer #1).

The reviewers have discussed the reviews with one another and the Reviewing Editor has drafted this decision to help you prepare a revised submission. In sum, the reviewers were positive about the experiments that address how neurofascin appears in the axon initial segment.

We would like to draw your attention to changes in our revision policy that we have made in response to COVID-19 (https://elifesciences.org/articles/57162). Specifically, we are asking editors to accept without delay manuscripts, like yours, that they judge can stand as *eLife* papers without additional data, even if they feel that they would make the manuscript stronger. Thus the revisions requested below mainly address clarity, presentation and interpretation.

Summary:

This study addresses the unresolved question of the cellular route for accumulation of neurofascin and Kv7.3 at axon initial segments. The authors use imaging approaches and neurofascin constructs fused to SEP or Dendra to study how neurofascin is delivered to the neuronal plasma membrane and the axon initial segment. The experiments support a model where both neurofascin and Kv7.3 are initially delivered to the somatic and axon terminal plasma membranes, and subsequently randomly diffuse through the membrane bilayer to the axon initial segment where they are trapped through binding to ankyrin-G. These are very interesting and overall well executed experiments. Attention to several questions and some issues of data analysis and presentation should be addressed and discussed.

Essential revisions:

1) The authors report accumulation of neurofascin and Kv7.3 at the cell body and nerve termini with apparently zero delivery to axons. It is possible that axonal proteins do exist but are below the detection limit. Given the large surface area of axons, even a small level could be significant. They should provide a lower limit of detection in axons, as well as estimates of total axonal neurofascin and Kv7.3 based on axonal membrane surface area.

2) The physical basis for low levels of exocytosis of neurofascin and Kv7.3 in axons should be discussed. One possibility is that exocytic vesicles are prevented from reaching the plasma membrane by the periodic spectrin-actin-adducin membrane skeleton

3) The rate of diffusion of SEP-neurofascin and Kv7.3 should be estimated to place dynamics of these proteins in context with those of other membrane proteins and lipids.

4) The authors validate SEP-Nfasc186 using neurons null for neurofascin and immunostaining for endogenous neurofascin. However, it would be important to demonstrate that neurofascin is indeed transported anterogradely to the axon terminals before its exocytosis at the terminals. Indeed, no data are shown in the axon shaft. Could it be that exocytic events are reduced in the axon itself given the reduced surface as compared to the cell body and axon terminals?

5) The authors show that the SEP construct diffuses along the axon but do not provide a mechanism by which this diffusion occurs. This suggest a passive mechanism but this is not discussed. Another possibility is that the SEP construct expression is too high and therefore the observed mechanism is a consequence of this overexpression. This is particularly obvious in Video 4 in which the level of SEP is high both at the AIS and the surrounding regions. There is a concern about overexpression, but this can be difficult to address. Analysis of ectopically expressed neurofascin-SEP regarding endogenous neurofascin could help.

6) In the control and FLIP experiments, it would be of great interest to follow the retrograde diffusion of SEP-Nfasc186 and its accumulation at the AIS upon neuronal maturation. Is neurofascin (and the SEP construct) still present at the axon terminals when neurons form synapses? Figure 3C-D, Could the author compare accumulation at the AIS versus more distal parts including axon terminals in mature circuits? Figure 3E, the authors should also show the axon terminal of the same neuron.

---

## [Author Response]

Essential revisions:1) The authors report accumulation of neurofascin and Kv7.3 at the cell body and nerve termini with apparently zero delivery to axons. It is possible that axonal proteins do exist but are below the detection limit. Given the large surface area of axons, even a small level could be significant. They should provide a lower limit of detection in axons, as well as estimates of total axonal neurofascin and Kv7.3 based on axonal membrane surface area.

SEP-Nfasc186 and Kv7.3 are not below the lower limit of detection in the axon. A line scan of SEP-Nfasc186 signal intensity at the cell soma, axon and axon terminal of the neuron in the upper panel of Figure 1C, now included as Figure 1D, shows that fluorescence was readily detectable in the axonal membrane relative to background. A line scan of SEP-Nfasc186 signal intensity in a Neurofascin-null cortical neuron shows the same (Figure 1—figure supplement 1B). Furthermore, a line scan now included as Figure 3—figure supplement 2B shows that the signal intensity of SEP-Kv7.3 in the axon is also readily detectable above background.

Therefore, the fact that TIRF microscopy does not detect significant numbers of exocytic fusion events per unit area in the axon is unlikely to be due to an inability to detect the protein at the axonal surface.

2) The physical basis for low levels of exocytosis of neurofascin and Kv7.3 in axons should be discussed. One possibility is that exocytic vesicles are prevented from reaching the plasma membrane by the periodic spectrin-actin-adducin membrane skeleton

Why a host of axonal vesicles, such as synaptic vesicles for example, ignore the axonal membrane on their way to fuse at the axon terminal is still imperfectly understood. The role of the axonal cytoskeleton in this context cannot be excluded. Therefore, we agree that this should be commented on. The text has been amended.

3) The rate of diffusion of SEP-neurofascin and Kv7.3 should be estimated to place dynamics of these proteins in context with those of other membrane proteins and lipids.

FRAP recovery data for Nfasc186 in the distal axon were fitted using a one dimension diffusion model (Ellenberg and Lippincott-Schwartz, 1999). The estimated diffusion coefficient for Nfasc186 is 0.37 ± 0.01 µm^2^/s and is comparable to a previously reported value of 0.34 ± 0.02 µm^2^/s for highly mobile, untethered axonal Nfasc186 in peripheral axons (Zhang et al., 2012). This information is now added to the manuscript.

4) The authors validate SEP-Nfasc186 using neurons null for neurofascin and immunostaining for endogenous neurofascin. However, it would be important to demonstrate that neurofascin is indeed transported anterogradely to the axon terminals before its exocytosis at the terminals. Indeed, no data are shown in the axon shaft. Could it be that exocytic events are reduced in the axon itself given the reduced surface as compared to the cell body and axon terminals?

We now include a video of vesicles transporting Nfasc186-mCherry showing abundant anterograde movement in the axon together with quantitation of their speed of movement (Video 4). Further, the data shown in Figure 3—figure supplement 1B, which shows that vesicles carrying Nfasc186-Dendra2 are transported anterogradely, were obtained in the distal axon immediately proximal to the axon terminal. This point is now made clearer in the text.

5) The authors show that the SEP construct diffuses along the axon but do not provide a mechanism by which this diffusion occurs. This suggest a passive mechanism but this is not discussed. Another possibility is that the SEP construct expression is too high and therefore the observed mechanism is a consequence of this overexpression. This is particularly obvious in Video 4 in which the level of SEP is high both at the AIS and the surrounding regions. There is a concern about overexpression, but this can be difficult to address. Analysis of ectopically expressed neurofascin-SEP regarding endogenous neurofascin could help.

Since neither the microtubule nor the actin-myosin systems contribute to the movement of SEP-Nfasc186 in the axonal membrane, we conclude that movement is indeed by passive diffusion (see point 3).

Video 4 (now Video 5) does not visualise the protein outside the AIS. The whole field of view comprises the AIS; there are no surrounding axonal regions visualised. We have clarified this point in the legend to Video 5 and Figure 4. We have addressed the issue of over-expression as follows:

i) FRAP-FLIP experiments on SEP-Nfasc186 expressed in Neurofascin-null neurons yielded identical results to those using WT neurons

ii) Endogenous neuronal Neurofascin is concentrated at the soma and axon terminal as found for SEP-Nfasc186

iii) Enrichment of SEP-Nfasc186 at the AIS relative to the soma and distal axon in WT neurons is as found in Neurofascin-null neurons, and is also as found for endogenous Neurofascin in the absence of SEP-Nfasc186.

These data support the view that SEP-Nfasc186 is a reliable surrogate for the behaviour of endogenous neuronal Nfasc186.

6) In the control and FLIP experiments, it would be of great interest to follow the retrograde diffusion of SEP-Nfasc186 and its accumulation at the AIS upon neuronal maturation. Is neurofascin (and the SEP construct) still present at the axon terminals when neurons form synapses? Figure 3C-D, Could the author compare accumulation at the AIS versus more distal parts including axon terminals in mature circuits? Figure 3E, the authors should also show the axon terminal of the same neuron.

As we now further emphasise in the manuscript, until it is trapped in the AIS the diffusion of SEP-Nfasc186 in the axonal membrane is bidirectional. Hence, after brief photoconversion of Nfasc186-Dendra2 in the axon terminal the signal becomes increasingly diluted, which in turn means that the signal from the converted protein in more proximal regions of the axon is below the detection limit.

The traffic of Neurofascin in mature neurons in vivo will be the subject of future work.

Figure 4—figure supplement 1, now included, shows enrichment of SEP-Nfasc186 at the AIS and axonal terminal of the same neuron at an early stage of AIS assembly.